# Postoperative Complications following Open Reduction and Rigid Internal Fixation of Mandibular Condylar Fracture Using the High Perimandibular Approach

**DOI:** 10.3390/healthcare11091294

**Published:** 2023-04-30

**Authors:** Hiroto Tatsumi, Yuhei Matsuda, Erina Toda, Tatsuo Okui, Satoe Okuma, Takahiro Kanno

**Affiliations:** 1Department of Oral and Maxillofacial Surgery, Faculty of Medicine, Shimane University, Izumo 693-8501, Japan; 2Department of Oral and Maxillofacial Surgery, National Hospital Organization Hamada Medical Center, Hamada 697-8511, Japan

**Keywords:** condylar fracture, high perimandibular approach, open reduction, rigid internal fixation, mouth opening

## Abstract

The high perimandibular approach is a feasible surgical technique for treating mandibular condylar fractures with open reduction and internal fixation, followed by fewer complications. Temporary trismus is the only postoperative complication that may occur. This study evaluated postoperative complications following open reduction and rigid internal fixation (OR-IF) of mandibular condylar fractures using the high perimandibular approach. Twenty consecutive patients undergoing OR-IF were included in this study. They included 11 male and 9 female patients, of an average age of 58.5 years, all of whom responded to a follow-up call at least 12 months after the surgery. All patients were evaluated for range of mouth opening, postoperative complications, and radiological findings. A statistical analysis of the relationship between range of mouth opening and related clinical parameters at 6 months postoperative evaluation was conducted. The fracture of the condylar neck was associated with a limited range of mouth opening and longer operation time. However, longer operation time was not associated with a limited range of mouth opening. The high perimandibular approach with OR-IF in mandibular condylar fractures is a feasible and safe technique; however, prolonged surgery and mandibular condylar neck fractures could affect the postoperative range of mouth opening.

## 1. Introduction

The mandibular condylar process is among the most frequent fracture sites, accounting for about 19–52% of mandibular fractures [1,2,3,4,5]. The crucial aspects in the treatment of condylar fractures include the restoration of mandibular ramus height, occlusion, facial asymmetry, and jaw function [6]. Open reduction and internal fixation (OR-IF) for condylar fractures is favorable and provides satisfactory clinical and early functional outcomes [7,8,9]. Nonetheless, there are reports that show that patients treated with closed reduction have relatively acceptable clinical and psychological outcomes [10]. As such, the type of treatment used for this fracture is controversial [11,12,13].

OR-IF is a difficult surgical procedure owing to the complex issues of stable osteosynthesis and requires a safe approach to the mandibular condylar fracture treatment. There is no standardized method for mandibular condylar fracture; thus, several surgical approaches have been used, including the preauricular, submandibular, retromandibular, and intra-oral approaches with or without endoscopic assists [14]. Kanno et al. have reported that postoperative facial nerve palsy could occur in some cases as a small percentage in the submandibular gland fascia and retromandibular transparotid approach techniques; however, no critical complications were mentioned [15,16]. Access to the fracture line is sometimes unsatisfactory in the most popular approaches, such as the Risdon or retromandibular technique, and especially in cases of high fracture as upper neck because the skin incision is made far from the fracture, and soft tissue retraction is difficult [17]. The high perimandibular approach/modified Risdon approach has been proposed as an innovative novel method to overcome the disadvantages of these previous standard approaches, such as the submandibular/retromandibular approaches, without compromising their advantages [18,19], and has been widely applied in clinical practice in recent years [20,21,22].

The high perimandibular approach is also known as the high submandibular approach [23], modified Risdon–Strasbourg approach [20,24], modified submandibular access [25], modified high submandibular approach [26,27], etc. In this technique, the marginal mandibular branch is not located within the retracted flap because of an initial superficial dissection of the platysma [18,19,28]. Therefore, access to the condyle is reportedly satisfactory, even in cases of high fracture, and the safety of the facial nerve is ensured [17,28]. A systematic review of surgical approaches for mandibular condylar fractures indicated that the high perimandibular approach is the safest in protecting the facial nerve in open treatment for subcondylar fractures [29].

The submandibular approach requires strong traction of the skin at the incision site, which may cause postoperative scarring due to contusion of the skin tissue at the wound margin [30]. The high perimandibular approach is a modified submandibular approach. This approach has the risk of postoperative scarring; however, scar formation is less noticeable behind the inferior margin of the mandible, and intraoperative skin traction is minimized, which results in less scarring and greater patient satisfaction [23,31]. Other postoperative complications, such as plate breakage and infection of the surgical field, have been reported [32], but are rare. The use of OR-IF in mandibular condylar fractures is a feasible and safe technique, even though there is a risk of inducing temporary postoperative trismus [26].

This study aimed to retrospectively evaluate postoperative complications and range of mouth opening following OR-IF of mandibular condylar fractures using the high perimandibular approach.

## 2. Patients and Methods

### 2.1. Study Design

The authors designed and implemented a retrospective cohort study and enrolled a sample derived from the population of patients who presented to the Maxillofacial Trauma Center, Shimane University Hospital (Shimane, Japan), between June 2019 and June 2021 for evaluation and management of condylar fractures.

This study was approved by the Ethics Committee of Shimane University Faculty of Medicine (approval number 20221006-1).

### 2.2. Patients

The inclusion criteria were (1) extracapsular mandibular condyle fracture for which surgical treatment of the neck or subcondylar regions consisted of the high perimandibular approach with rigid internal fixation, with complete medical records available for evaluation by the authors; (2) availability of preoperative and postoperative panoramic radiographs or computed tomographic (CT) images; (3) mental status permitting an adequate neuromotor examination; and (4) regular postoperative follow-up over 12 months, documented on clinical and radiographic evaluation charts.

Patients who did not meet the inclusion criteria (e.g., treated by other surgical approaches such as the endoscopically assisted transoral approach) and those who did not have regular follow-up evaluation for over 12 months were excluded.

### 2.3. Evaluated Variables

In this study, the patients’ profiles (age and gender), mechanism of injury, site of mandibular condylar fracture (condylar neck or condylar base according to the classification of the AO Foundation [33]), type of fracture (deviation, displacement, or dislocation, according to the classification of MacLennan [34]), presence of associated maxillofacial fractures, and operation time (only the time required to reduce the mandibular condyle fracture was obtained from the anesthesia record) were recorded. The range of mouth opening at 6 months postoperatively, postoperative complications (surgical site infection, facial nerve palsy, surgical scar perceptibility, malocclusion, and TMJ pain), radiological evaluations (state of reduction, breakage of plates, and screw loosening), and factors that influence the range of mouth opening were evaluated. The extent of mouth opening was measured using a ruler to determine the distance between the upper and lower incisors when the patient opened the mouth to its natural maximum [10]. The exercises were performed under the observation of a medical professional to avoid stress on the surgical site and sutures. For pain assessment, the oral surgeon’s question was, “Do you have jaw pain?” The same question was asked by the patient to standardize the assessment of pain.

### 2.4. Surgical Procedures

All patients were treated under general anesthesia with transnasal intubation by a single surgical team at the Maxillofacial Trauma Center. Some patients had associated midfacial and mandibular fractures. In such cases, they were first reduced and fixed rigidly with maxillomandibular fixation (MMF) in centric occlusion using an inter-maxillary fixation (IMF) screw with 0.3 mm steel wire or dental arch bars in cases involving dental or alveolar trauma, followed by OR-IF of the associated fracture site.

In the high perimandibular approach, a 4–5 cm incision was made 0.5 cm below the inferior border of the mandible to include the mandibular angle (Figure 1a). The skin was incised to the level of the platysma muscle and then undermined approximately 2 cm upward over the platysma muscle. The platysma muscle was incised approximately 1 cm parallel to the inferior border of the mandible (Figure 1b). The masseter muscle fascia was then cut until the masseter muscle was exposed, taking care not to damage the facial nerve (Figure 1c). The masseter was incised above the border of the lower mandible, and the periosteum was dissected. The masseter muscle was stripped as high as possible from the mandible along with the periosteum to expose the fracture site (Figure 1d). The fractured condylar segment was then reduced anatomically by inferior distraction of the mandibular ramus. Condylar fracture fixation was performed using the double-plate technique for stabilization. In this technique, a 2.0 mm titanium plate was first fixed to the posterior border buttress of the mandibular ramus; this was followed by a second fixation with the same plate at the anterior buttress of the condyle in the same manner (Figure 1e). No additional surgical incisions were required for instrumentation. Wound closure was performed after confirming occlusion, perfusion of the wound, and hemostasis (Figure 1f). Intermaxillary fixation was not performed postoperatively; however, elastic control training was conducted.

### 2.5. Rehabilitation after Operation

All patients were instructed to open their mouths manually and spontaneously to the maximum range of opening after surgery. Opening exercises were performed by an oral surgeon and dental hygienist on an outpatient basis at least once per day. The maximum mouth opening was where the patient felt mild pain and was instructed to maintain that position for 10 s. The exercise was performed 12 times during one rehabilitation session, with rests of 60 s every three sets [35]. The oral surgeon and dental hygienist instructed the patients to perform the exercises by themselves not only during hospitalization but also for six months after discharge, when their mouth opening was reevaluated. If the patients’ mouth opening had decreased after six months, they were instructed to continue the mouth opening exercises. All home exercises were performed manually, and no special device was used. The dental hygienist who instructed the patient in oral opening exercises was a staff member who had been with the practice for at least three years and had extensive knowledge of the mandibular condylar fracture. The instructional methods among the staff members were also calibrated within the department.

### 2.6. Statistical Analysis

For descriptive statistics, the median (interquartile range (IQR)) was calculated. The relationship between range of mouth opening and clinical parameters (age and operation time) and between operation time and age were analyzed using the Pearson product–moment correlation coefficient. The Mann–Whitney U test and Kruskal–Wallis test were used for group comparisons between range of mouth opening and clinical parameters (sex, cause of trauma, site of fracture, situation of fracture, and presence of associated trauma) and between operation time and same clinical parameters.

All statistical analyses were performed using EZR (Saitama Medical Center, Jichi Medical University, Saitama, Japan), a graphical user interface for R (The R Foundation for Statistical Computing, Vienna, Austria). More precisely, EZR is a modified version of the R commander designed to add statistical functions frequently used in biostatistics. Statistical significance was set at *p* < 0.05.

## 3. Results

Patient profiles, categorization of mandibular condylar fractures, type of fracture, association of maxillofacial fractures, total operation time, and cause of injuries are summarized in Table 1.

Twenty patients were included in the study. Fifteen patients were injured due to falls, two due to motor vehicle accidents, and one each due to sports, violence, and work. In 10 cases, the fracture sites were located in the neck and subcondylar regions. Twelve sites showed deviation, five showed displacement, and the remaining three showed dislocation of the fracture type. Isolated fractures of the mandibular condyle were observed in 7 patients and mandibular, maxilla, or zygoma fractures in 13 patients. Of these, two patients had additional fractures of the condylar head on the other side.

Operation time, range of mouth opening, postoperative complications (surgical site infection, facial nerve palsy, surgical scar perceptibility, malocclusion, temporomandibular joint pain), and radiological evaluations (poor state of reduction, breakage of plates, and screw loosening) are summarized in Table 2.

The median operation time was 55.0 (47.25–77.25) min. All patients were followed up for 1 year, and surgical infection was not observed immediately after surgery. In some cases, the buccal branch of the facial nerve was exposed in the surgical field; however, there were no cases of postoperative facial nerve palsy (Figure 1g). Skin scarring was minimal, almost invisible, and not particularly problematic at postoperative 6 months in all patients (Figure 1h). No malocclusion or TMJ pain was observed in any of the cases 1 year postoperatively.

The range of mouth opening was >35 mm in 17 patients, with a median of 42.0 mm (Figure 1i). Three patients had mild trismus but did not complain of feeding disorders. Radiological evaluations revealed anatomical reduction with good bone union and no evidence of plate and/or screw breakage or loosening in any case (Figure 1j).

A significant difference was found between the range of mouth opening and fracture sites, and a correlation trend was found between the fracture site; however, no significant differences were observed in the other factors (Figure 2 and Table 3). Furthermore, a significant difference was found only between operative time and fracture site (Figure 3, Table 4).

## 4. Discussion

The major finding of our study is that the high perimandibular approach is a safe surgical procedure that is not affected by the patient’s advanced age or fracture status. The advantages of this high perimandibular approach are that it is not influenced by the patient’s background, has a wide range of indications, and is not affected by the treatment of other concomitant maxillofacial fractures. Therefore, a high perimandibular approach is an excellent and simple surgical technique with a low risk of complications that does not rely on the surgeon’s years of experience. Although safety and complication rates tend to be the focus of surgical procedure evaluation, it is also crucial that the procedure can be easily performed by a larger number of surgeons, and the high perimandibular approach is a beneficial procedure that meets these criteria.

In the high perimandibular approach, a skin incision is designed just below and parallel to the edge of the mandibular angle, with a shorter working distance from the skin incision to the mandibular condyle. Furthermore, the incision line of the platysma and masseter muscles can be designed upward from the marginal mandibular branch of the facial nerve, thereby protecting it. Therefore, this technique is recommended to provide good access to the surgical field and to expand the working space [18,19]. Similarly, in the retromandibular transparotid approach for the fractured condylar neck region, the advantages include a shorter working distance from the skin incision to the condyle and better access to the sigmoid notch on the posterior border of the mandible, with direct visual alignment of the fractured segments [16,20,36]. The retromandibular approach requires incision of the parotid capsule and blunt dissection of the parotid gland toward the posterior border of the mandible. As a result, salivary fistulae frequently occur as a postoperative complication [37,38]; our previous report showed that it has an occurrence rate of 1 in 19 cases [16]. In contrast, the high perimandibular approach performed in this study allowed for easier access to the fracture site because it exposed the condyle with less tissue and did not pass through other organs.

Mandibular condylar fracture surgeries using the extraoral approach are frequently associated with facial nerve palsy. According to Lutz et al. [31], the marginal mandibular branch is combined with other facial nerve branches at a rate of 0–16%. Several studies have reported that the incidence of facial nerve disturbance is 0–0.9% with the high perimandibular approach [20,23]. Thus, this technique is feasible in preserving the marginal mandibular branch of the facial nerve, where the incision line can be designed upward from the inferior border branch of the facial nerve. In our previous study, the submandibular approach had a 4.2% facial nerve disturbance rate [15], while a 11% facial nerve disturbance rate was observed with the retromandibular approach, whereas an 11% facial nerve disturbance rate was observed with the retromandibular approach [9]. In the present study, no postoperative complications of nerve palsy were observed. The main difference between the traditional submandibular (Risdon) approach and high cervical transmasseteric anteroparotid/high perimandibular approaches is the superoinferior levels of the supraplatysmal dissection and subsequent deepening of the condyle. While traditional submandibular approaches through the area include subplatysmal dissection and run transversely under the marginal mandibular branch, high cervical transmasseteric anteroparotid/high perimandibular approaches dissect the supraplatysma and run through the upper layer. The superiorly retracted flap is smaller than the traditional submandibular approach, allowing for easier reduction of the fractured condyle and decreasing the risk of facial nerve injury [39].

These techniques, as well as the transmasseteric anteroparotid approach [23,30], involving an incision through the masseter muscle to reach the fracture site, pose the risk of trismus due to postoperative contracture. Pau et al. [26] have noted that the high perimandibular approach involves masseter muscle dissection, which can cause temporary postoperative trismus. Nam et al. [24] have reported a mean postoperative range of mouth opening of 46.6 mm. Nowair et al. [20] have revealed that mouth opening in group I reached normal values (>40 mm) in 11 patients, of whom only one patient had a slight impairment (38 mm); no postoperative aperture impairment was observed. Other reports have shown that the average postoperative range of mouth opening was 42 mm at 3 months [31] and 49 mm at 6 months [23], which is generally similar to other extraoral approaches [36,40]. Therefore, postoperative trismus does not appear to be a problem with this technique. Further, our study showed good results, with a median mouth opening of 42 mm. However, three patients in the present study had a range of mouth opening ≤35 mm. This study evaluated the factors influencing the range of mouth opening. The results showed a significant difference in fracture site and correlation trends in operation time. Prolonged surgery and mandibular condylar neck fractures probably increase the amount of incision, traction, and traction time of the masseter muscle to obtain the operative field. Therefore, contusion of the masseter muscle was induced, which may have affected the postoperative range of mouth opening. In addition, it was suggested that there may be an association between the extent of opening and operative time; however, the number of cases in this study was small (20 cases). The pathology and severity of the fracture may have a strong confounding effect; however, the small number of cases did not allow for multivariate analysis. That is, it is unlikely that the surgeon’s years of experience affected the operative time or the amount of opening. For standardization, the direction of opening was limited to the vertical direction, and measurements were not taken in lateral or thrusting motions.

Operation time is a cause of surgical site infection and other postoperative complications [41,42,43]. Therefore, this study further compared operative time with clinical parameters. The results showed that the neck of operative time was significantly longer than subcondyle at site of fracture. This suggests that it may be more difficult to reach the mandibular condylar neck in this technique than in the subcondyle. For mandibular condylar neck fractures, choosing a safer and easier to reach technique, such as the transmasseteric anteroparotid approach, may be warranted [29,44,45].

Patients with facial trauma are generally more likely to be young males [1], and compared to other studies [20,24], the mean age of our patients was 58.5 years. Therefore, low postoperative activity and inadequate functional training may cause trismus. To prevent postoperative contracture of the masseter muscle, several studies have reported that incision of the masseter muscle should be parallel to the muscle bundle [32,46]. Therefore, these techniques should be considered for indications in older patients. Moreover, we should compare the outcomes of these 20 patients with sex/age matching controls who underwent surgery with the submandibular or endoscopically assisted transoral approach as a future study. In addition, the safety and efficacy of the high perimandibular approach should be further verified in future studies using multiple outcomes. As an example, this study evaluated only the vertical distance between incisors. In addition to this evaluation, it will be necessary in future studies to evaluate three-dimensional mandible movements, including distance of horizontal movement and mandibular protrusion [47]. Further, it is necessary to consider a research design that includes occlusal force and masticatory function as outcomes as an ideal and ultimate goal.

This study had two limitations. First, the amount of opening was only evaluated 6 months postoperatively. Second, only 20 patients were included in the study. Nevertheless, the difference in the type of fracture was not found to have any effect on the range of mouth opening. More cases and more continuous evaluations are required in future studies.

## 5. Conclusions

This technique provides a reliable and safe clinical approach that reduces major postoperative complications, except for the possibility of postoperative trismus. If the potential for postoperative trismus can be improved, the high perimandibular approach should replace other extraoral approaches as the predominant surgical approach for mandibular condylar fractures at the site of the neck and subcondyle.

## Figures and Tables

**Figure 1 healthcare-11-01294-f001:**
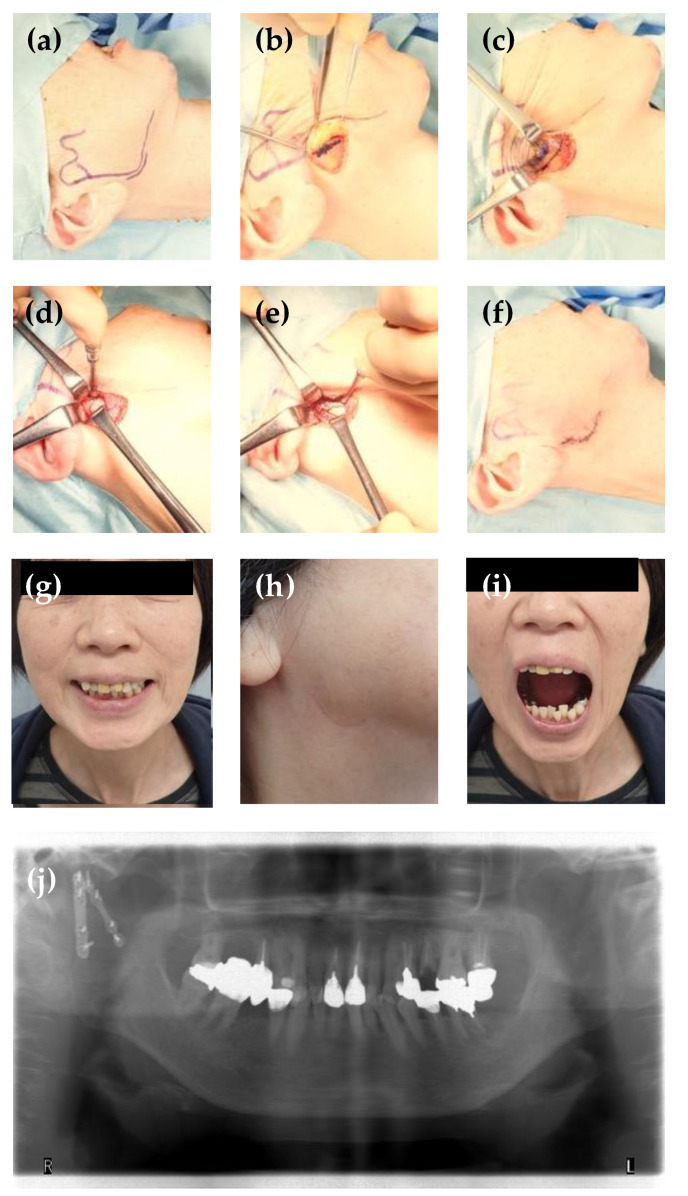
A 69-year-old woman presented with a fracture to the right lower neck with deviation caused by a fall. (**a**) Incision design for high perimandibular approach. (**b**) Incision marked for platysma muscle, after undermining approximately 2 cm upward over the it. (**c**) Incision marked for the masseter muscle. (**d**) The masseter muscle was dissected and fracture site exposed. (**e**) Reduction of fractured condylar fragment, and rigid internal fixation using two locking miniplates. (**f**) After skin closure. (**g**) Photograph 6 months after surgery showing no motor nerve disturbance of the facial nerve. (**h**) Skin scarring was minimal 6 months postoperatively. (**i**) Photograph showing mouth-opening capacity at 6 months postoperative. (**j**) Six months postoperative orthopantomogram view showing proper fracture reduction and rigid fixation.

**Figure 2 healthcare-11-01294-f002:**
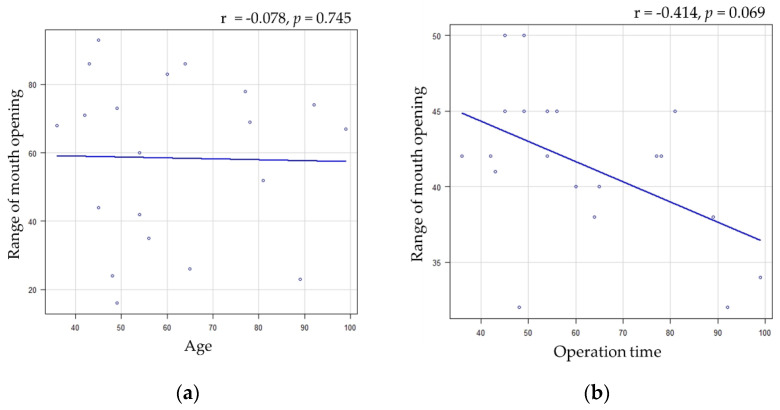
Relationship between range of mouth opening and related factors: (**a**) relationship between range of mouth opening and age and (**b**) relationship between range of mouth opening and operation time.

**Figure 3 healthcare-11-01294-f003:**
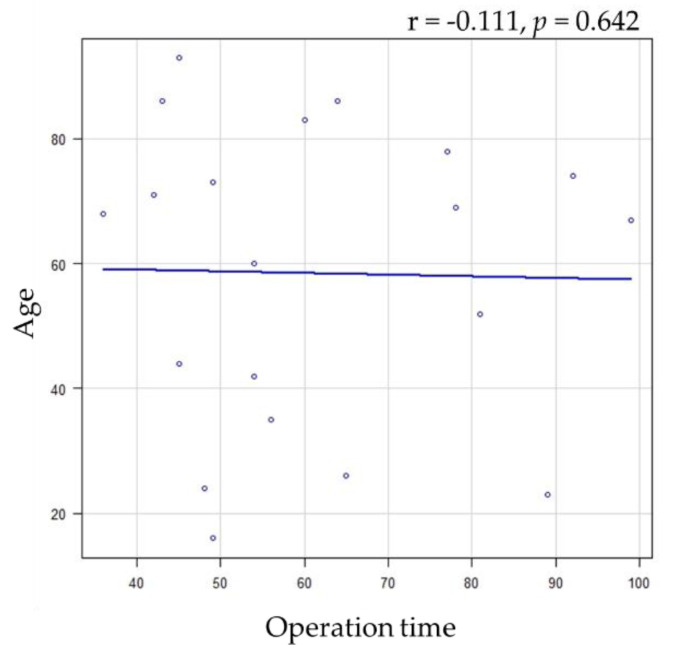
Relationship between operation time and age.

**Table 1 healthcare-11-01294-t001:** Demographic characteristics of the patients who underwent OR-IF.

Item	Category	N (%) or Median (25–75 Percentile)
Sex	Male	11 (55.0)
	Female	9 (45.0)
Age		67.5 (40.3–75.0)
Cause of injury	Fall	15 (75.0)
	Traffic accident	2 (10.0)
	Sports	1 (5.0)
	Violence	1 (5.0
	Work	1 (5.0)
Site of fracture	Neck	10 (50.0)
	Subcondyle	10 (50.0)
Type of fracture	Deviation	12 (60.0)
	Displacement	5 (25.0)
	Dislocation	3 (15.0)
Presence of associated maxillofacial fractures	Yes	13 (65.0)

N: number of patients.

**Table 2 healthcare-11-01294-t002:** Postoperative outcomes.

Item	Category	N (%) or Median (25–75 Percentile)
Operation time (only condyle fracture)		55.0 (47.25–77.25)
Range of mouth opening		42.0 (39.5–45.0)
Surgical site infection	Yes	0 (0)
Facial nerve palsy	Yes	0 (0)
Surgical scar perceptibility	Yes	0 (0)
Malocclusion	Yes	0 (0)
Temporomandibular joint pain	Yes	0 (0)
Poor state of reduction	Yes	0 (0)
Plate breakage	Yes	0 (0)
Screw loosening	Yes	0 (0)

N: number of patients.

**Table 3 healthcare-11-01294-t003:** Group comparison between range of mouth opening and clinical parameters.

Item	Category	Range of Mouth Opening (Median (25–75 Percentile))	*p* Value
Sex	Male (*n* = 11)	45.0 (42.0–45.0)	0.15
	Female (*n* = 9)	41.0 (40.0–42.0)	
Cause of trauma	Fall (*n* = 15)	42.0 (39.0–45.0)	0.64
	Traffic accident (*n* = 2)	42.0 (42.0–42.0)	
	Others (*n* = 3)	45.0 (41.5–47.5)	
Site of fracture	Neck (*n* = 10)	39.0 (35.0–42.0)	0.02 *
	Subcondyle (*n* = 10)	43.5 (42.0–45.0)	
Type of fracture	Deviation (*n* = 12)	41.5 (40.0–45.0)	0.38
	Displacement (*n* = 5)	38.0 (38.0–42.0)	
	Dislocation (*n* = 3)	45.0 (43.5–45.0)	
Presence of associated maxillofacial fractures	Yes (*n* = 13)	42.0 (40.0–45.0)	0.31
	None (*n* = 7)	42.0 (38.0–42.0)	

*: significant difference, N: number of patients.

**Table 4 healthcare-11-01294-t004:** Group comparison between operation time and clinical parameters.

Item	Category	Operation Time (Median (25–75 Percentile))	*p* Value
Sex	Male (*n* = 11)	54.0 (48.5–79.5)	0.88
	Female (*n* = 9)	60.0 (45.0–65.0)	
Cause of trauma	Fall (*n* = 15)	60.0 (51.5–77.5)	0.68
	Traffic accident (*n* = 2)	39.0 (37.5–40.5)	
	Others (*n* = 3)	49.0 (47.0–69.0)	
Site of fracture	Neck (*n* = 10)	70.5 (55.5–87.0)	0.03 *
	Subcondyle (*n* = 10)	49.0 (43.5–55.5)	
Type of fracture	Deviation (*n* = 12)	49.0 (44.5–57.0)	0.08
	Displacement (*n* = 5)	78.0 (64.0–89.0)	
	Dislocation (*n* = 3)	77.0 (65.5–79.0)	
Presence of associated maxillofacial fractures	Yes (*n* = 13)	54.0 (45.0–65.0)	0.38
	None (*n* = 7)	64.0 (51.5–83.5)	

*: significant difference, N: number of patients.

## Data Availability

All data have been illustrated in the manuscript.

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
