# Peer review of "Postoperative Complications following Open Reduction and Rigid Internal Fixation of Mandibular Condylar Fracture Using the High Perimandibular Approach"

_healthcare, 2023, doi:10.3390/healthcare11091294_

Round 1

Reviewer 1 Report

The manuscript describes a retrospective cohort study evaluating postoperative complications and the range of mouth opening following open reduction and rigid internal fixation of mandibular condylar fractures using a "new" approach (the high perimandibular approach).

The subject is interesting but the manuscript has much to improve, as detailed in the attached document. Moreover, professional proofreading of English writing is mandatory.

Author Response

Responses to Comments/Suggestions

from Reviewer 1

Dear reviewer 1

We are truly grateful to your critical comments and thoughtful suggestions for our manuscript, which have helped us improve our manuscript further. Based on these comments and suggestions, we have carefully revised our original manuscript. All changes made to the main text are presented in red font. Please find our point-by-point responses to your comments/questions below.

Sincerely,

Takahiro Kanno,

(article: healthcare-2265343)

Comments to the Author

Title

  1. There are two grammatical errors in the Title. Please, replace it with: Postoperative complications following open reduction and rigid internal fixation of mandibular condylar fracture using the high perimandibular approach.

Response: Thank you very much for your comment. As you pointed out, we have changed the title to “Postoperative complications following open reduction and rigid internal fixation of mandibular condylar fracture using the high perimandibular approach”.

Abstract:

  1. There are some grammatical mistakes. The entire manuscript should be carefully revised by an English native speaker.

Response: Thank you for your comment. The revised manuscript has been carefully checked by an English native speaker. The certificates for English revisions are attached here.

  1. “Results: The patients did not develop facial nerve palsy, surgical site infection, or other conditions. Limited range of mouth opening and fracture site of condylar neck and longer operation time were statistically significant clinical factors (p < 0.05).”

You should rewrite it. The fracture of the condylar neck was associated with a limited range of mouth opening and with higher operation time. Higher operation time was NOT associated with a limited range of mouth opening.

Response: Thank you for your comment. As you pointed out, we have modified the sentence “The patients did not develop facial nerve palsy, surgical site infection, or other conditions. Limited range of mouth opening and fracture site of condylar neck and longer operation time were statistically significant clinical factors (p < 0.05).” to “The fracture of the condylar neck was associated with a limited range of mouth opening and higher operation time. Higher operation time was not associated with a limited range of mouth opening.”.

  1. “Conclusion: The high perimandibular approach with OR-IF in mandibular condylar fractures is a feasible and safe technique; however, prolonged surgery and mandibular condylar neck fractures could affect the postoperative range of mouth opening.”.
  2. Again, is conclusion is not in accordance with the findings. You did not achieve a statistically significant association between the range of mouth opening and operation time (P = 0.069)

Response: Thank you for your comment. As you pointed out, we did not get a statistical significant result; hence, we modified the relevant text as follows: “The high perimandibular approach with OR-IF in mandibular condylar fractures is a feasible and safe technique; however, prolonged surgery and mandibular condylar neck fractures could affect the postoperative range of mouth opening.”.

  1. Keywords: Please, format them according to the journal style.

Response: Thank you very much for your comment. As you pointed out, we have revised carefully the keywords according to the journal style.

  1. “Our previous study showed the feasibility of the submandibular approach using the submandibular gland fascia and retromandibular transparotid approach techniques, post-operative facial nerve palsy could occur in some cases as a small percentage, but a critical complication [8,9]”.

The entire sentence is strange, it seems like some punctuations and information are missing. Moreover, avoid self-citations like this (“our previous study”) in the Introduction. The phrase must be reformulated.

Response: Thank you for your comment. As you pointed out, we have carefully checked and modified the relevant text as follows: “Our previous study showed the feasibility of the submandibular approach using the submandibular gland fascia and retromandibular transparotid approach techniques, post-operative facial nerve palsy could occur in some cases as a small percentage, but a critical complication [8,9]” to “Kanno et al. have reported that postoperative facial nerve palsy could occur in some cases as a small percentage in the submandibular gland fascia and retromandibular transparotid approach techniques, but a critical complication almost was not mentioned.”

  1. “The high perimandibular approach is also known as the high submandibular approach [14],modified Risdon–Strasbourg approach [13,15], modified submandibular access [16], modified high submandibular approach [17,18] etc.”

If you are reporting all possible names of this technique, do not use ‘etc’, report them all.

Response: Thank you very much for your comment. As you pointed out, we have removed “etc” as we do not intend to showcase all technologies.

  1. The authors stated that “The high perimandibular approach is a modified submandibular approach”; therefore, would be possible to compare the outcomes of these 20 patients with gender/age matching controls that underwent surgery using the submandibular approach or even the endoscopically assisted transoral approach?

Response: Thank you for your comment. As you mentioned, we should compare the outcomes of these 20 patients with gender/age matching controls who underwent surgery using the submandibular approach or endoscopically assisted transoral approach. However, we are unable to compare the patients in this study because the method of comparison you mentioned is not included in the current study protocol. Therefore, we have added it as a suggestion for future study at the end of the discussion.

  1. How the range of mouth opening was evaluated? With a regular rule? With a digital caliper rule? With a kinesiograph? Please, clarify it in the text.

Response: Thank you for your comment. As you pointed out, we have added a more detailed measurement of range of mouth opening in “2.3. Evaluated Variables”.

  1. The photos of Figure 1 should be grouped, three pictures per line, and the fourth line will be the panoramic radiography in a bigger picture.

Response: Thank you for your comment. As you pointed out, we have made corrections to Figure 1.

  1. The pictures of the face should be edited like the example below:

Response: Thank you for your comment. As you pointed out, we have made corrections to (g), (h), and (i) in Figure 1.

  1. Please, provide high-quality images.

Response: Thank you for your comment. As you pointed out, we replaced it with an uncompressed image file.

  1. “The Mann–Whitney U test and Kruskal–Wallis test were used for group comparisons”. In this study, you do not have more than one group. Please, remove the term ‘group’

Response: Thank you for your comment. In this study, we conducted a two- or three-group comparison, as shown in Tables 3 and 4. In the group comparison, the Mann–Whitney U-test and Kruskal–Wallis test were used. Therefore, this sentence has been retained.

  1. The text below Table 1 is just repeating what was already reported in the table. Please, rewrite it.

Response: Thank you for your comment. The content of this text provides more detailed information not included in the table. Therefore, it was unmodified.

  1. “The median operation time was 55.0 (36–99) minutes”. It is not in accordance with Table 2.

Response: Thank you for your comment. As you pointed out, we have modified the median (quartile range).

  1. Figures 2 and 3 should be deleted, since they could be reported in Tables 2 and 3, respectively.

Response: Thank you for your comment. As you pointed out, we did not remove the scatter plots, as we believe the figures will help readers visually understand the scatter of the data.

  1. Moreover, report the ‘r’ value for all parameters in the Tables.

Discussion:

Response: Thank you for your comment. Since the analysis in the table is a group comparison using the Mann–Whitney U-test and Kruskal–Wallis test, it is not possible to show correlation coefficients. Therefore, the tables were not modified as such.

  1. The first paragraph should report your main findings and, from that point, start discussing them.

Response: Thank you for your comment. As you pointed out, the first paragraph of the discussion should have been the major finding of our study. Therefore, we have added a paragraph.

  1. Your Discussion seems a literature review more than a section to discuss what you have found.

Response: Thank you for your comment. As you noted, we have discussed the findings from the study in the first paragraph.

  1. Please, carefully revise and rewrite it making more comparisons related to your results.

Response: Thank you for your comment. As you pointed out, we believe that it was particularly crucial to discuss whether the complications of the procedure depend on the surgeon’s years of experience. Therefore, we have added this information to the Discussion section.

Conclusion:

  1. You do not need to state the aim of this study again, you just need to ‘answer’ it. Be as direct as possible.

Response: Thank you for your comment. As you pointed out, we have removed the purpose section in the Conclusions section.

  1. “Informed Consent Statement: Not applicable.” If you are evaluating human beings, you need to have an informed consent form signed by all volunteers.

Response: Thank you for your comment. We have obtained permission from the Ethics Review Board to conduct this study and have obtained consent from all patients, which we have noted.

Reviewer 2 Report

I would like to acknowledge the authors for their valuable work on " Postoperative complication following open reduction and rigid internal fixation of mandibular condylar fracture using high perimandibular approach". The manuscript is written well; few minor corrections need to be done before publication which I mentioned in sticky note in the PDF file. Kindly find the attachment for correction.

Best of luck.

Author Response

Responses to Comments/Suggestions

from Reviewer 2

Dear reviewer 2

We are truly grateful to your critical comments and thoughtful suggestions for our manuscript, which have helped us improve our manuscript further. Based on these comments and suggestions, we have carefully revised our original manuscript. All changes made to the main text are presented in red font. Please find our point-by-point responses to your comments/questions below.

Sincerely,

Takahiro Kanno,

(article: healthcare-2265343)

Comments to the Author

Abstract should be not sturctural. Kindly check the format of the journal.

Response: Thank you for this suggestion. As you pointed out, we have changed the abstract to one paragraph without structural forms.

Please remove the numbers before the keywords.

Response: Thank you for this suggestion. As you pointed out, we have removed the numbers before the keywords.

Kindly rephrase this sentence.

We have rephrased the sentence you pointed out for the readers to understand it more easily.

The

Response: Thank you for this suggestion. As you pointed out, we have added “T” to this word.

Check the spacing.

Response: Thank you for this suggestion. We have modified the spacing in the section you pointed out.

This was abbreviated before, so no need to use the full form.

Response: Thank you for this suggestion. As you pointed out, we have used the abbreviation.

Modify the table title please. only Demographic characteristics is not a complete sentence. And also remove the (N=20) from the title of the table. Also add the footnote for the table. For example, abbreviation of N, %

Same comments as Table 1.

Response: Thank you for this suggestion. As you pointed out, we have modified the title of Table 1 and removed the number of patients (N=20). Further, we have added table footnotes.

Please consistent with either ’N’ or ’n’.

Response: Thank you for this suggestion. We used “N” and “n” interchangeably for the number of patients. We used “N” for the entire target population and “n” for the number of subjects divided into groups. This rule is the same rule used in other articles in this journal. Therefore, we have maintained this notation.

No need to cite same reference in adjacent two sentences. rephrase the two sentences and cite the reference once.

Response: Thank you for this suggestion. As you pointed out, we have removed the forward citations and retained the backward citations.

Open

Response: Thank you for this suggestion. We have removed the sentence according to a suggestion by another reviewer.

Reviewer 3 Report

The presented research aimed to evaluate postoperative complications following open reduction and rigid internal fixation of mandibular condylar fractures using the high perimandibular approach. The results highlighted that the patients did not develop facial nerve palsy, surgical site infection, or other conditions and that the limited range of mouth opening, fracture site of the condylar neck, and longer operation time were statistically significant clinical factors. The manuscript is well structured, with the main purpose so clear. 

However, I have several important suggestions. 

MAJOR:

Patients and Methods 

- line 204: range of mouth 102 opening - I suggest adding in addition to the MMO results also the range of lateral movements and protrusion. Furthermore, please define whether it is an active or passive movement. Also, add whether it was painless or painful - and whether these measurements were standardized across participants.

- Line 104: and TMJ pain – did you also consider the pain in the masticatory muscles or just the pain in the temporomandibular joint?

OVERALL:

Moreover, the work needs to contain information on the methodology of measuring the range of mouth opening (was it repeated several times for normalization? etc.) The work also lacks information on whether the patients had any physiotherapy treatments or whether there was rehabilitation after the procedure. It may also be important to know what TMJ range of motion patients had before the procedure, as it could significantly affect the result of the observation. All these elements require explanation and discussion and should be included in the work (or in the section on work limitations).

MINOR:

Please correct the general work style, punctuation, stylistic errors, and gramma (e.g., line 50)

In general, the work is interesting and can contribute to the literature. However, many deficiencies in the presented work must be addressed in the paper before publication. I hope my suggestions will help improve this work.

Author Response

Responses to Comments/Suggestions

from Reviewer 3

Dear reviewer 3

We are truly grateful to your critical comments and thoughtful suggestions for our manuscript, which have helped us improve our manuscript further. Based on these comments and suggestions, we have carefully revised our original manuscript. All changes made to the main text are presented in red font. Please find our point-by-point responses to your comments/questions below.

Sincerely,

Takahiro Kanno,

(article: healthcare-2265343)

Comments to the Author

Title

The presented research aimed to evaluate postoperative complications following open reduction and rigid internal fixation of mandibular condylar fractures using the high perimandibular approach. The results highlighted that the patients did not develop facial nerve palsy, surgical site infection, or other conditions and that the limited range of mouth opening, fracture site of the condylar neck, and longer operation time were statistically significant clinical factors. The manuscript is well structured, with the main purpose so clear. However, I have several important suggestions.

Response: Thank you for taking the time to read our manuscript. We have carefully responded to your questions.

MAJOR:

Patients and Methods

- line 204: range of mouth 102 opening - I suggest adding in addition to the MMO results also the range of lateral movements and protrusion. Furthermore, please define whether it is an active or passive movement. Also, add whether it was painless or painful - and whether these measurements were standardized across participants.

Response: Thank you for your comment. As you pointed out, more detailed definitions for measuring range of mouth opening have been added to “2.3. Evaluated Variables”. When indicating the range of mouth opening, the patient was instructed to make a vertical opening, and the measurement was taken. Therefore, we did not obtain data on lateral or protrusion movements; hence, we could not add them. Moreover, all ranges of mouth openings were passive movements of the patient; thus, this was also added. In the current study, the presence or absence of pain was standardized by collecting data from patients using the same questions.

- Line 104: and TMJ pain – did you also consider the pain in the masticatory muscles or just the pain in the temporomandibular joint?

Response: Thank you for your comment. The evaluation method (questions to the patient) for pain in this case did not identify whether the pain was in the temporomandibular joint or masticatory muscles. Therefore, we did not add a note to that effect.

OVERALL:

Moreover, the work needs to contain information on the methodology of measuring the range of mouth opening (was it repeated several times for normalization? etc.) The work also lacks information on whether the patients had any physiotherapy treatments or whether there was rehabilitation after the procedure. It may also be important to know what TMJ range of motion patients had before the procedure, as it could significantly affect the result of the observation. All these elements require explanation and discussion and should be included in the work (or in the section on work limitations).

Response: Thank you for your comment. As you pointed out, we have added the sentence on physiotherapy and rehabilitation of mouth opening after surgery in “2.5 Rehabilitation after surgery”.

MINOR:

Please correct the general work style, punctuation, stylistic errors, and gramma (e.g., line 50)

Response: Thank you for your comment. As you pointed out, we have checked and corrected the general work style, punctuation, stylistic errors, and grammar.

In general, the work is interesting and can contribute to the literature. However, many deficiencies in the presented work must be addressed in the paper before publication. I hope my suggestions will help improve this work.

Response: We believe that your suggestions and comments have helped improve the quality of our manuscript. Thank you very much.

Round 2

Reviewer 3 Report

Dear Authors,

Thank you for considering my comments. In its current form, the work meets the criteria for publication in the journal. Congratulations, and good luck with your further research.